# SEFAR: SPARSE-FEATURE-BASED REGULARIZATION FOR FINE-TUNING ON LIMITED DOWNSTREAM DATA

## ABSTRACT

A commonly employed approach within the domain of transfer learning is fine-tuning the meticulous crafting of novel loss functions or the subtle adjustment of either all or a part of the parameters in the pre-trained network. However, most of the current fine-tuning methods typically require a substantial amount of downstream data, which can be limiting in real-world scenarios. When dealing with limited data, an appropriate regularization method can be used to enhance a model's generalization capabilities and reduce the risk of overfitting. In this paper, we propose a SparsE-FeAture-based Regularization (SEFAR) method that can significantly enhance the performance of any fine-tuning method when there is a limited amount of downstream data available. Our proposed method is simple to implement: it leverages the results generated by sparse features to self-distill the results produced by complete features. This paper also provides insight into how the SEFAR works: one is a relation to the generalization bound of a kernel regression problem, and the other is the flatness of the minima. Additionally, extensive empirical experiments demonstrate the benefits of this method for fine-tuning on various datasets using different backbones. The code will be released soon.

## 1 INTRODUCTION

Due to the release of large-scale datasets like ImageNet (Krizhevsky et al., 2012), fine-tuning has become a popular approach in the field of transfer learning, which updates some or all of the parameters of a pre-trained network on downstream data or tasks. Previous methods have focused on designing specific loss functions or identifying the most suitable parameters for updating, thereby enhancing the fine-tuning performance. Most of these papers are based on the default assumption that sufficient images are available for fine-tuning, which means their experiments are conducted with training on the entire downstream training dataset. However, we claim that this assumption is not reasonable in some practical scenarios in which the downstream images are hard to collect, such as satellite image recognition, diagnosis of skin diseases. In this paper, we concentrate on a more realistic scenario: fine-tuning using a limited set of downstream images. To mitigate the risk of overfitting on these constrained images, we introduce SparsE-FeAture-based Regularization (SEFAR) and incorporate it into existing fine-tuning techniques to improve their performance.

Figure 1 illustrates the fundamental idea of SEFAR: While retaining the original fine-tuning approach, a sparse binary mask is utilized to obtain a sparse version of the original features. Subsequently, using these sparse features, a task-specific head is employed to compute the same task loss function. At the same time, we use the predictions generated from these sparse features to distill the predictions generated from the complete features. The additional classification and self-distillation loss function computed using the sparse features together constitute our proposed SEFAR, which serves as a regularization technique for the original fine-tuning process.

Through experiments and theory, we have demonstrated the feasibility and effectiveness of SEFAR as a regularization method. From the theoretical perspective, SEFAR essentially provides independent regularization for the original features $\mathbf{F}$ and the sparse features $\mathbf{K}$. SEFAR can be regarded as an approach that operates within a balanced framework, optimizing both the Rademacher-complexity-based regularizer and the isometry of the sparse features. The learning of the sparse features will assist in regularizing the label information. Regarding the experiment, we find that

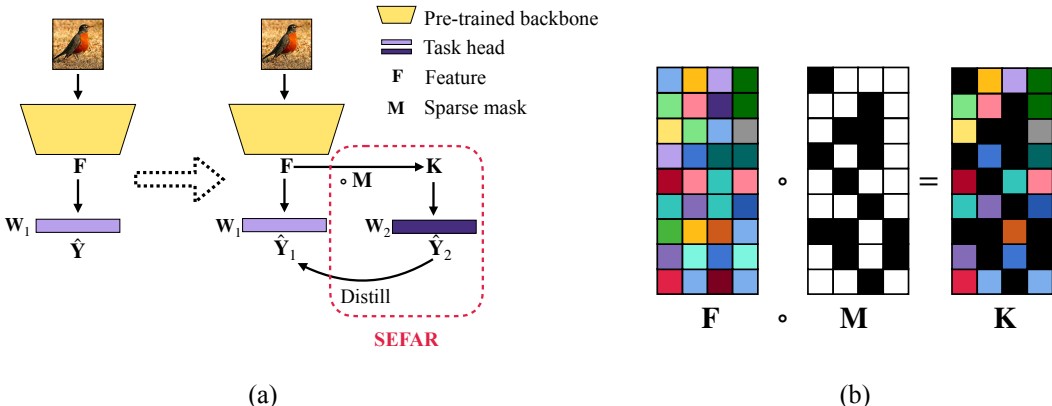

Figure 1: (a) The fundamental principle of SEFAR: While retaining the original network and loss function, a new classification head, $\mathbf{W}_2$, is introduced. It takes a sparse feature matrix $\mathbf{K}$ ($\mathbf{K} = \mathbf{F} \circ \mathbf{M}$, where $\mathbf{M}$ is a sparse binary mask) as the input to compute the classification loss. Furthermore, the classification results generated by $\mathbf{W}_2$ are utilized to concurrently distill those produced by the original features $\mathbf{F}$. (b) The schematic diagram illustrating the generation of sparse feature $\mathbf{K}$.

when the training sample size is limited, whether in the traditional fine-tuning paradigm or in the meta-learning-based fine-tuning paradigm, SEFAR can enhance the classification accuracy of various fine-tuning methods. Furthermore, SEFAR exhibits a beneficial impact across different datasets and with various backbone architectures. This validates its efficacy as a regularization method that can function as a plug-and-play module, seamlessly integrated into any fine-tuning method to enhance performance when insufficient training images are available. We have explored why SEFAR can improve the classification performance: Introducing SEFAR not only automatically optimizes the upper bound of generalization error during the training process but also makes the model to converge to the region with a flatter loss.

The contributions of our paper are as follows:

- In response to the challenge of limited fine-tuning data in real-world scenarios, we introduce SEFAR, a straightforward yet effective regularization method. It can serve as a plug-and-play module, seamlessly integrated into various fine-tuning approaches, thereby enhancing their classification performance.

- We provide some insight into the mechanism of how the SEFAR works well. First, we show a theoretical insight that one can regard SEFAR as a method for learning a data-dependent kernel, thereby reducing the upper bound of a certain generalization error. Second, we empirically show that the SEFAR prefers flat minima.

- We experimentally demonstrate that SEFAR can be used across different datasets, backbones, and fine-tuning methods to consistently improve the classification performance while fine-tuning with insufficient images.

## 2 RELATED WORK

**Fine-tuning** With the release of large-scale datasets, pretraining–fine-tuning has become a commonly used technique in transfer learning and exhibits formidable capabilities in downstream tasks. To address the issue of catastrophic forgetting, $L^2$-SP (Xuhong et al., 2018) employs pretrained model parameters as constraints. By introducing an L2 norm between the pretrained model parameters and the fine-tuned model parameters, it brings the pretrained model's parameters closer to those of the original model, thus enabling the network to retain its prior knowledge. Kirkpatrick et al. (2017) introduced elastic weight consolidation (EWC). When EWC is applied in fine-tuning, it becomes L2-SP-Fisher, as it computes the Fisher matrix of the network weights. Instead of parameters, DELTA (Li et al., 2019) constrains the behavior of the corresponding layer, specifically, the features generated by that layer itself. Batch spectral shrinkage (BSS), proposed by Chen et al.

(2019), constrains the singular values in the network features. Song et al. (2021) proposed a hybrid forward network for alternately updating the layer weights of the student model. It takes into account the dynamic balance between knowledge transfer losses and task-specific losses during training. TransTailor (Liu et al., 2021) not only updates the weights of the pretrained model but also modifies the network's structure by pruning and fine-tuning the pretrained model based on the importance of target-aware weights. Inspired by FitNets (Romero et al., 2014), L2T (Jang et al., 2019) explores which layers between the original network and the fine-tuned network should be aligned for knowledge transfer as well as which features and how much knowledge from each layer should be transferred. Spottune (Guo et al., 2019) introduces a policy network to make routing decisions, determining whether to forward images to the fine-tuning or pretrained layers. Adafilter (Guo et al., 2020b) employs an LSTM network to selectively fine-tune convolutional filters based on the activations from the previous layer, optimizing them individually for each example. Based on the gradient distance, Wan et al. (2019) proposed that models can be fine-tuned more effectively if we can constrain the difference between cross-entropy loss and the L2 gradient. Co-tuning (You et al., 2020) constrains the mapping between different semantic spaces of the pretrained domain and the downstream task domain. Surgical tuning (Lee et al., 2022) selectively fine-tunes different layers in the network for various types of domain gaps.

## 3 SPARSE-FEATURE-BASED REGULARIZATION (SEFAR)

We assume that the loss function $L_1$ represents the loss function of a particular fine-tuning method. The original features generated by the backbone are denoted as $\mathbf{F}$, while the sparse version of these features is denoted as $\mathbf{K}$, where $\mathbf{K} = \mathbf{F} \circ \mathbf{M}$. $\mathbf{M}$ is a sparse binary mask. $\mathbf{W}_1$ is the classification head for computing $L_1$ with $\mathbf{F}$ and $\mathbf{W}_2$ is for computing SEFAR with $\mathbf{K}$ and $\mathbf{F}$. $\hat{\mathbf{Y}}_1 = \mathbf{W}_1\mathbf{F}$ and $\hat{\mathbf{Y}}_2 = \mathbf{W}_2\mathbf{K}$ represent the computed results of two classification heads, respectively.

Our SEFAR is a plug-and-play regularization method that can be applied to any fine-tuning method. In $L_1$, we calculate all the losses associated with the original fine-tuning method, such as cross-entropy loss and another regularization term. SEFAR consists of two components: $L_2$ and the distillation loss $L_3$ from $\hat{\mathbf{Y}}_2$ to $\hat{\mathbf{Y}}_1$. $L_2$ represents the cross-entropy loss between $\hat{\mathbf{Y}}_2$ and the ground truth $\mathbf{Y}$. The distillation loss $L_3$ is calculated as follows:

$$L_3 = \text{KL}(\hat{\mathbf{Y}}_1/t, \hat{\mathbf{Y}}_2^*/t), \tag{1}$$

where $\hat{\mathbf{Y}}_2^*$ denotes the gradient-detached version of $\hat{\mathbf{Y}}_2$, and $t$ is the temperature coefficient. So, the total loss function $L$ can be expressed as follows:

$$L = L_1 + L_2 + \lambda L_3, \tag{2}$$

where $\lambda$ is a hyperparameter to balance the different loss components. Please note that self-distillation $L_3$ here does not necessarily have to be measured using KL divergence; other distance metrics, such as mean squared error (MSE) loss, can also be used. This has also been confirmed through experiments presented in Section 4.2. The pipeline is represented in Algorithm 1.

## 4 EXPERIMENTS

### 4.1 RESULTS OF FINE-TUNING METHODS

*Conventional fine-tuning regime* We utilize a pretrained ResNet-18 model on ImageNet-1K as the backbone, conducting experiments on five datasets, namely EuroSAT (Helber et al., 2019), CropDisease (Mohanty et al., 2016), ISIC (Codella et al., 2019), ChestX (Wang et al., 2017), and CIFAR-10, with and without SEFAR. Various fine-tuning techniques are applied to show the effectiveness of SEFAR, including Baseline (optimizing all network parameters by minimizing the cross-entropy loss), linear probing (keeping the backbone parameters fixed and only updating the parameters of the linear classifier), $L^2$-SP (Xuhong et al., 2018), DELTA (Li et al., 2019), and surgical fine-tuning (Lee et al., 2022).

We first randomly select 80% of each dataset for model testing. From the remaining 20%, we completely randomly sample 200 images for model training without considering the number of classes in the datasets. To ensure the credibility of our results, we conduct ten experiments with different

---

Algorithm 1: Pipeline of Fine-tuning with SEFAR

---

**Input:** Images: $\mathbf{X}$, Label: $\mathbf{Y}$
**Ouput:** A model fine-tuned with SEFAR
$\mathbf{F}$: Feature generated by the backbone
$m(\cdot)$: A pre-trained backbone
$\mathbf{W}_1$: Classification head 1
$\mathbf{W}_2$: Classification head 2
$d_f$: The dimensionality of a feature vector
maskgen$(d_1, d_2)$: Returning a sparse binary mask with the shape of $(d_1, d_2)$
**for** $\mathbf{X}$, $\mathbf{Y}$ in dataloader **do**
 $\quad \mathbf{M} \leftarrow$ maskgen($\mathbf{X}$.shape[0], $d_f$)
 $\quad \mathbf{F} \leftarrow m(\mathbf{X})$
 $\quad \mathbf{K} \leftarrow \mathbf{F} \circ \mathbf{M}$
 $\quad L \leftarrow L_1(\mathbf{W}_1\mathbf{F}, \mathbf{Y}) + L_2(\mathbf{W}_2\mathbf{K}, \mathbf{Y}) + \lambda L_3(\mathbf{W}_1\mathbf{F}, \mathbf{W}_2\mathbf{K}.detach())$
 $\quad$ Update the network
**end for**

---

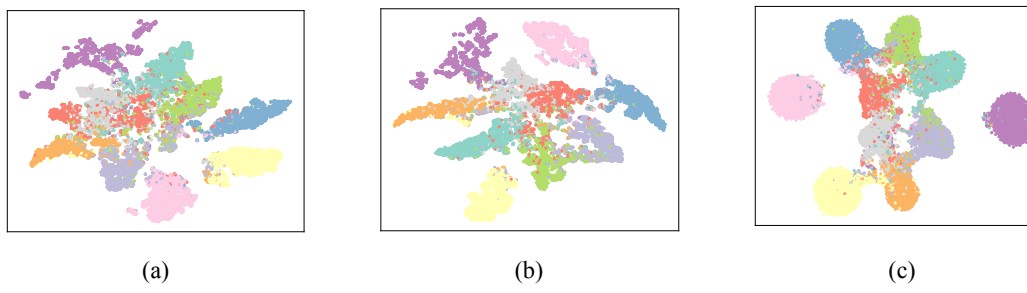

(a)              (b)              (c)

Figure 2: t-SNE results of (a) Baseline w/o SEFAR's feature $\mathbf{F}$, (b) Baseline w/ SEFAR's original feature $\mathbf{F}$ and (c) Baseline w/ SEFAR's sparse feature $\hat{\mathbf{F}}$ on the testing dataset of EuroSAT. It is evident that the sparse feature clustering exhibits better performance, making it a suitable choice as the teacher for distillation. This also implies that, in scenarios with limited data for fine-tuning, randomly dropping some dimensions to generate sparse features may result in enhanced discriminative capabilities.

random seeds and reported the final average accuracy. In each set of control experiments, with and without SEFAR, we use the same hyperparameters, including random seeds (from 0 to 9), batch size (200), learning rate (0.001), image size (224), and optimizer (Adam). In other words, all settings are identical except for the presence or absence of SEFAR. To ensure the model can be fine-tuned adequately, we train the network for 500 epochs in each experiment. Unless otherwise specified, all subsequent experiments are conducted using similar settings. The performance can be found in Table 1, and SEFAR can consistently improve classification accuracy across different datasets and fine-tuning methods by up to 3.42%. Baseline w/ SEFAR's t-SNE Van der Maaten & Hinton (2008) visualization of $\mathbf{F}$ and $\mathbf{K}$ is shown in Figure 2. Even though, theoretically, SEFAR primarily provides regularization for the features $\mathbf{F}$ and $\mathbf{K}$ themselves, in the context of linear probing experiments where only fixed features are generated, we observe a positive impact of SEFAR. We attribute this to the fact that the random masks introduced by SEFAR also effectively apply dropout to classification head $\mathbf{W}_2$. Consequently, learning $\mathbf{W}_2$ indirectly influences the learning of classification head $\mathbf{W}_1$, leading to improved classification performance.

To investigate the impact of dropout introduced by SEFAR on the enhancement of linear probing, we conduct experiments using random binary masks $\mathbf{M}$ with varying levels of sparsity and different distillation temperatures on the EuroSAT dataset, as illustrated in Figure 3. We observe that within a substantial range of hyperparameter values (Sparsity: 0.1~0.7, Temperature: 0.5~10, Weight: 0.1~10), SEFAR consistently improves the fine-tuning performance of linear probing.

*n-way k-shot fine-tuning manner* Besides the conventional fine-tuning manner, we also conduct experiments with the *n*-way *k*-shot manner, which is often studied in relation to the few-shot prob-

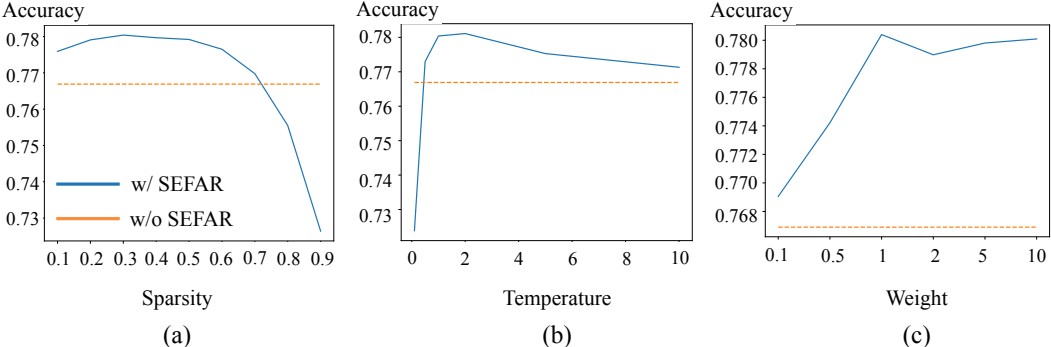

Figure 3: Classification accuracy of linear probing with different hyperparameters on the EuroSAT dataset. (a) **Different sparsity** values of random binary masks with a temperature coefficient of 1 and weight of 1. (b) **Different temperature** values with a sparsity value of 0.3 and weight of 1. (c) **Different weight** values with a sparsity value of 0.3 and temperature of 1. The dashed line represents the performance without using SEFAR. The results indicate that, even from a theoretical perspective, SEFAR introduces regularization to feature learning. The dropout introduced by SEFAR in linear probing consistently improves fine-tuning performance within a substantial range of hyperparameter values.

lem (Vinyals et al., 2016; Snell et al., 2017). Specifically, we randomly build 600 episodes. We randomly select $n = 5$ classes in each episode and then chose $k = 1\ or\ 5$ samples from each class as the support set for fine-tuning the model. Subsequently, we select $c$ samples from each class to construct the query set for calculating the classification accuracy. On the support set in each episode, baseline and linear probing are applied to tune the ResNet-10 pretrained on Mini-ImageNet (Guo et al., 2020a) using the SGD optimizer with an initial learning rate of 0.01. The performance is represented by the average classification accuracy over the 600 episodes within the 95% confidence interval. The performance can be seen in Table 2. Except for the 5-shot experiment on the CropDisease dataset, SEFAR can improve accuracy in all other groups.

## 4.2 ABLATION STUDY

In this section, we first address three important questions: *a*) whether each component of SEFAR has a positive impact, *b*) whether other distance functions can be used instead of KL divergence in the distillation loss, and *c*) whether SEFAR can be applied while fine-tuning Vision Transformer models.

To verify *a*), we explore the impact of $L_2$ and $L_3$ on the results. To validate *b*), we replace the KL divergence in $L_3$ with MSE loss. To verify *c*), we introduce the ViT-S as the backbone. All validation experiments are conducted based on the Baseline method. The performance is presented in Table 4. The table illustrates that both $L_2$ and $L_3$ are crucial in our method. In addition, MSE loss can be utilized as the self-distillation term as well. Moreover, SEFAR is not confined to CNN-based networks but can also be applied to Vision Transformers. These results demonstrate SEFAR's excellent scalability.

We also verify the classification performance of the original classification head $\mathbf{W}_1$ and the additional classification head $\mathbf{W}_2$ introduced by SEFAR, as shown in Table 3. Obviously, when using SEFAR, both with $\mathbf{W}_1$ and $\mathbf{W}_2$, there is an improvement in classification performance compared to not using SEFAR. Furthermore, in most cases, the random masking has a minimal impact on $\mathbf{W}_2$.

## 4.3 WHY DOES SEFAR GENERALIZE BETTER?

Section 4.3.1 has theoretically demonstrated that SEFAR can regularize the learning of features. In this section, we will illustrate how SEFAR enhances the generalization of fine-tuning methods from the perspective of changes in the upper bound of the generalization error and flatness.

|  |  | EuroSAT | ISIC | CropDisease | ChestX | CIFAR-10 |
|---|---|---|---|---|---|---|
| Base | w/o SEFAR | 83.55 | 70.85 | 68.69 | 36.72 | 55.65 |
|  | w/ SEFAR | 86.05 | 72.23 | 71.61 | 38.37 | 58.06 |
|  | Δ | **+2.5** | **+1.38** | **+2.92** | **+1.65** | **+2.41** |
| linear probing | w/o SEFAR | 76.69 | 69.76 | 64.07 | 35.74 | 63.35 |
|  | w/ SEFAR | 78.04 | 70.07 | 64.51 | 36.60 | 63.98 |
|  | Δ | **+1.35** | **+0.31** | **+0.44** | **+0.86** | **+0.63** |
| $L^2$-SP | w/o SEFAR | 83.21 | 71.00 | 63.43 | 38.45 | 60.08 |
|  | w/ SEFAR | 85.29 | 71.70 | 66.85 | 39.05 | 62.78 |
|  | Δ | **+2.08** | **+0.70** | **+3.42** | **+0.60** | **+2.70** |
| DELTA | w/o SEFAR | 88.93 | 70.96 | 72.22 | 37.32 | 68.46 |
|  | w/ SEFAR | 90.22 | 72.29 | 74.10 | 38.19 | 70.05 |
|  | Δ | **+1.29** | **+1.33** | **+1.88** | **+0.87** | **+1.59** |
| Surgical Finetuning | w/o SEFAR | 86.78 | 71.55 | 68.64 | 36.79 | 61.00 |
|  | w/ SEFAR | 87.49 | 72.30 | 72.08 | 37.66 | 62.57 |
|  | Δ | **+0.71** | **+0.75** | **+3.44** | **+0.87** | **+1.57** |

Table 1: The classification accuracy results (%) for five fine-tuning methods with and without SE-FAR. Δ represents the gain introduced by SEFAR. The table demonstrates that SEFAR consistently enhances classification performance across different fine-tuning methods and datasets (w/ SEFAR: with SEFAR, w/o SEFAR: without SEFAR). The hyperparameter configuration of w/ SEFAR is presented in Table 5 in the Appendix.

|  |  | EuroSAT | ISIC | CropDisease | ChestX |
|---|---|---|---|---|---|
| Baseline | w/o SEFAR | 51.82±0.84 | 31.53±0.56 | 60.93±0.93 | 21.66±0.36 |
|  | w/ SEFAR | 52.70±0.78 | 32.31±0.56 | 62.12±0.88 | 22.06±0.36 |
| linear probing | w/o SEFAR | 57.27±0.90 | 31.34±0.56 | 66.93±0.93 | 21.65±0.38 |
|  | w/ SEFAR | 59.97±0.85 | 31.97±0.61 | 68.22±0.85 | 21.99±0.39 |

|  |  | EuroSAT | ISIC | CropDisease | ChestX |
|---|---|---|---|---|---|
| Baseline | w/o SEFAR | 78.57±0.61 | 48.70±0.60 | 89.83±0.53 | 25.36±0.40 |
|  | w/ SEFAR | 79.74±0.57 | 49.43±0.61 | 89.42±0.54 | 25.73±0.41 |
| linear probing | w/o SEFAR | 77.74±0.61 | 43.80±0.58 | 89.71±0.54 | 24.88±0.40 |
|  | w/ SEFAR | 78.05±0.61 | 44.83±0.57 | 89.90±0.54 | 25.39±0.42 |

Table 2: Comparison of 5-way 1-shot (above) and 5-shot (below) experiments between w/ SEFAR and w/o SEFAR. This table demonstrates that SEFAR can improve the classification accuracy in meta-task fine-tuning as well. The hyperparameter configuration of w/ SEFAR is pre- sented in Table 6 in the Appendix.

| | w/o SEFAR | w/ SEFAR ($\mathbf{W}_1$) | w/ SEFAR ($\mathbf{W}_2$, w/ mask) | w/ SEFAR ($\mathbf{W}_2$, w/o mask) |
|---|---|---|---|---|
| Baseline | 83.55 | 86.05 | 85.45 | 85.77 |
| linear probing | 76.69 | 78.04 | 73.78 | 78.27 |
| $L^2$-SP | 83.21 | 85.29 | 85.40 | 86.00 |
| Delta | 88.93 | 90.22 | 90.11 | 90.20 |
| Surgical | 86.78 | 87.49 | 87.84 | 88.06 |

Table 3: Ablation study on which classification head works in SEFAR. w/ SEFAR ($W_1$) means incorporating SEFAR and testing it using the classification head $\mathbf{W}_1$. w/ SEFAR ($\mathbf{W}_2$, w/ mask) means incorporating SEFAR, using the classification head $\mathbf{W}_2$, and simultaneously applying random masking during testing. Similarly, w/ SEFAR ($\mathbf{W}_2$, w/o mask) represents testing without the random mask. The table demonstrates that $\mathbf{W}_1$ and $\mathbf{W}_2$ possess comparable classification abilities. Furthermore, even when introducing random masks during testing, $\mathbf{W}_2$ exhibits strong robustness.

### 4.3.1 SEFAR REDUCES THE UPPER BOUND OF GENERALIZATION ERROR

Next, we will provide a theoretical explanation for the effectiveness of SEFAR. For simplicity, suppose one-class (output) case with the MSE loss. The SEFAR loss is represented as follows:

$$L(\mathbf{w}_1, \mathbf{w}_2; \mathbf{F}, \mathbf{F}) = ||\mathbf{w}_1\mathbf{F} - \mathbf{y}||^2 + ||\mathbf{w}_2\mathbf{K} - \mathbf{y}||^2 + ||\mathbf{w}_1\mathbf{F} - \mathbf{s}||^2 + 2\lambda_1\|\mathbf{w}_1\|^2 + \lambda_2\|\mathbf{w}_2\|^2, \quad (3)$$

where $\mathbf{F}, \mathbf{K} \in \mathbb{R}^{d \times b}$, $\mathbf{w}_{1,2} \in \mathbb{R}^d$, and $\mathbf{K} = \mathbf{M} \circ \mathbf{F}$. $\mathbf{M}$ can be sparse. $\mathbf{s}$ is a constant vector. In the current case, detached $\mathbf{s}$ is equal to $\mathbf{w}_2\mathbf{K}$. Then, one can easily obtain the solution $\mathbf{w}_i^* := \operatorname{argmin}_{\mathbf{w}_i^*} L(\mathbf{w}_1, \mathbf{w}_2; f, k)$ with fixed features $(\mathbf{F}, \mathbf{K})$:

$$\mathbf{w}_1^* = (\mathbf{s} + \mathbf{y})(\mathbf{F}^\top\mathbf{F} + \lambda_1\mathbf{I})^{-1}\mathbf{F}^\top/2, \quad \mathbf{w}_2^* = \mathbf{y}(\mathbf{K}^\top\mathbf{K} + \lambda_2\mathbf{I})^{-1}\mathbf{K}^\top. \quad (4)$$

Then, substituting them back, we have

$$L(\mathbf{w}_1^*, \mathbf{w}_2^*; \mathbf{F}, \mathbf{K}) = \lambda_1 R_1(\mathbf{F}) + \lambda_2 R_2(\mathbf{K}) + const., \quad (5)$$

with

$$R_1(\mathbf{F}) := \frac{1}{2}(\mathbf{s} + \mathbf{y})^\top(\mathbf{F}^\top\mathbf{F} + \lambda_1\mathbf{I})^{-1}(\mathbf{s} + \mathbf{y}), \quad R_2(\mathbf{K}) = \mathbf{y}^\top(\mathbf{K}^\top\mathbf{K} + \lambda_2\mathbf{I})^{-1}\mathbf{y}. \quad (6)$$

This function form of $R_1(\mathbf{F})$ (and $R_2(\mathbf{K})$) is well known in the literature on kernel methods (Hu et al., 2020). If $\mathbf{K}^\top\mathbf{K}$ is a (untrained) positive-definite kernel function, $R(\mathbf{K})$ is an upper bound of the generalization error known as the Rademacher complexity. Thus, we can interpret that the SEFAR effectively regularizes these bounds through the training of $\mathbf{F}$ and $\mathbf{K}$. Note that we have the regularizers for $\mathbf{F}$ and $\mathbf{K}$, respectively. First, $R_2(\mathbf{K})$ enforces the features $\mathbf{K}$ to achieve data-dependent kernel minimizing of the generalization bound. Second, $R_1(\mathbf{F})$ enforces the features $\mathbf{F}$ to achieve the data-dependent kernel for self-distilled output $\mathbf{y} + \mathbf{s}$. Although the training makes the kernels dependent on data and we lose the rigorous meaning of the Rademacher complexity, it is still interesting that the SEFAR is regarded as an effective regularizer for the features through these bounds.

To interpret the interaction between head 2 ($\mathbf{K}$) and head 1 ($\mathbf{F}$), note that the detached $\mathbf{s}$ with $\mathbf{w}_2^*$ is given by $\mathbf{s}^* = \mathbf{y}(\dot{\mathbf{K}}^\top\dot{\mathbf{K}} + \lambda_2\mathbf{I}_B)^{-1}\dot{\mathbf{K}}^\top\dot{\mathbf{K}}$, where $\dot{\mathbf{K}}$ means detached. Substituting this $\mathbf{s}^*$ into $R_1(\mathbf{F})$, we can explicitly state the following:

$$R_1(\mathbf{F}) = \mathbf{y}^\top(2\mathbf{I} - \lambda_2(\dot{\mathbf{K}}^\top\dot{\mathbf{K}} + \lambda_2\mathbf{I})^{-1})(\mathbf{F}^\top\mathbf{F} + \lambda_1\mathbf{I})^{-1}(2\mathbf{I} - \lambda_2(\dot{\mathbf{K}}^\top\dot{\mathbf{K}} + \lambda_2\mathbf{I})^{-1})\mathbf{y}. \quad (7)$$

Denote the eigenvalue decomposition of $\mathbf{K}^\top\mathbf{K}$ by $\mathbf{K}^\top\mathbf{K} = \mathbf{U}^\top\mathbf{\Lambda}_k\mathbf{U}$. Considering Equation 2, we have the following:

$$2\mathbf{I} - \lambda_2(\dot{\mathbf{K}}^\top\dot{\mathbf{K}} + \lambda_2\mathbf{I})^{-1} =: \mathbf{U}^\top\mathbf{\Lambda}'\mathbf{U}, \quad (8)$$

where $(\mathbf{\Lambda}')_{ii} = 2 - \lambda_2/(\lambda_{k,ii} + \lambda_2)$. Here, for simplicity, assume that the eigenvalues of $\mathbf{K}^\top\mathbf{K}$ are divided into two components, $\lambda_{k,ii} \gg \lambda_2$ and $\lambda_{k,ii} \ll \lambda_2$, which is proven to be reasonable in Figure 4. We denote the set of the former indices by $\uparrow$ and the set of the latter ones by $\downarrow$. Then, we have the following:

$$\mathbf{U}^\top\mathbf{\Lambda}'\mathbf{U} \sim 2\mathbf{U}_\uparrow^\top\mathbf{U}_\uparrow + \mathbf{U}_\downarrow^\top\mathbf{U}_\downarrow = \mathbf{I} + \mathbf{U}_\uparrow^\top\mathbf{U}_\uparrow. \quad (9)$$

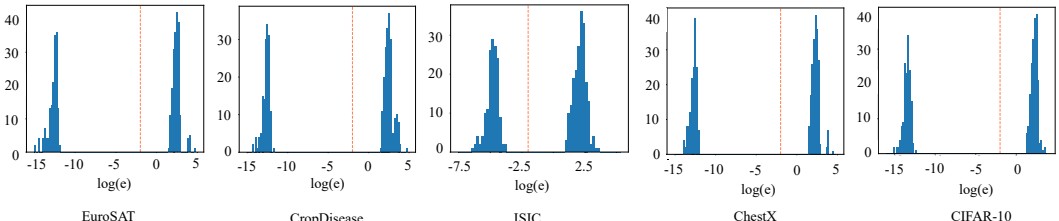

Figure 4: Eigenvalue distributions of $\mathbf{K}^\top\mathbf{K}$, where $\lambda_2$ is set to 0.01 and $\log(\lambda_2) = -2$ (the orange dotted line). The horizontal axis represents the logarithm (base 10) of the eigenvalue $e$. Because some of the eigenvalues $e$ are equal to zero, to avoid issues with logarithms, we manually filter them out. In each of the five datasets, there are two peaks in their distributions, and all of them are far from $\log(\lambda_2) = -2$.

Then, $R_1(\mathbf{F})$ becomes the following:

$$\mathbf{y}^\top(\mathbf{I} + \mathbf{U}_\uparrow^\top\mathbf{U}_\uparrow)(\mathbf{F}^\top\mathbf{F} + \lambda\mathbf{I})^{-1}(\mathbf{I} + \mathbf{U}_\uparrow^\top\mathbf{U}_\uparrow)\mathbf{y}. \tag{10}$$

If the kernel $\mathbf{F}^\top\mathbf{F}$ is fixed, one can regard this as the generalization bound with an "effective label" $\tilde{\mathbf{y}} := (\mathbf{I} + \mathbf{U}_\uparrow^\top\mathbf{U}_\uparrow)\mathbf{y}/2$. If the head 2 with the sparsity has a low-dimensional feature space denoted by $\uparrow$, we can interpret that the training of the feature $\mathbf{F}$ in head 1 tries to regularize the label information along with this $\uparrow$ space.

**Remark on Rademacher complexity for generalization.** Due to the ability of Rademacher complexity to assess the generalization capability of a class of machine learning models, it can be employed as an optimization objective to mitigate the issue of overfitting. Zhai & Wang (2018) showed that the Rademacher complexity is bounded by a function related to the dropout rate and used this bound as a regularizer. LocalDrop (Yousefi et al., 2018) represents a novel approach for regularizing neural networks based on local Rademacher complexity. In our SEFAR framework, through derivation, we also observe that SEFAR introduces terms based on Rademacher complexity into the overall loss function, allowing it to optimize the upper bound of the generalization error directly.

Formula 5 demonstrates that the introduction of SEFAR not only optimizes the overall loss function but also optimizes the upper bounds of both $\mathbf{F}$ and $\mathbf{K}$. Figure 6 presents the changes in upper bounds during the training process for both baseline with SEFAR and linear probing with SEFAR using Formulas 6.

To mitigate the influence of randomness, we conduct ten experiments and reported the average values. During the training of linear probing with SEFAR, the values of $R_1(\mathbf{F})$ and $R_2(\mathbf{K})$ remain relatively constant. This indicates that if the features are fixed, these upper bounds cannot be naturally optimized through the learning of the classification head. Conversely, in the training of Baseline w/ SEFAR, in which the backbone is trainable, $R_1(\mathbf{F})$ and $R_2(\mathbf{K})$ can be optimized. Based on the comprehensive analysis above, we conclude that the optimization introduced by SEFAR on the upper bound, along with the dropout in the classification head, collectively contribute to the improved fine-tuning performance of the network.

### 4.3.2 SEFAR MAKES THE MODEL CONVERGE TO THE REGION WITH FLATTER LOSS

We also give the one-dimensional visualization of flatness (Li et al., 2018) of Baseline w/ SEFAR and w/o SEFAR across all the datasets. Note that the flatness has been used in the literature of deep learning as a metric for measuring generalization performance (Keskar et al., 2016; Jiang et al., 2019). Figure 5 illustrates that, when the same perturbation is added to the model, the training loss curves of Baseline w/ SEFAR (blue curves) are lower and flatter compared to those of Baseline w/o SEFAR (orange curves). This indicates that SEFAR enhances the model's resistance to perturbations during the training process, making it more robust. This may also be one of the reasons why SEFAR contributes to better model generalization.

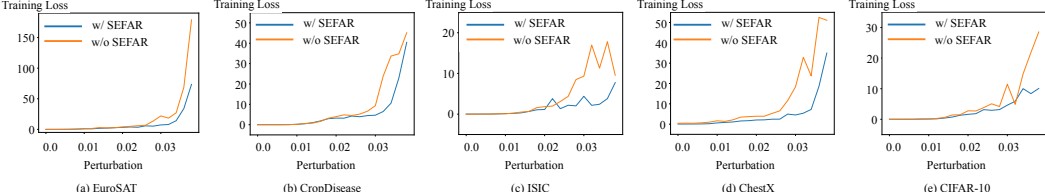

(a) EuroSAT  (b) CropDisease  (c) ISIC  (d) ChestX  (e) CIFAR-10

Figure 5: Comparison of training loss between Baseline with SEFAR (blue curves) and without SEFAR (orange curves) after adding the same perturbation across five datasets. All the curves of Baseline with SEFAR increase at a flatter rate than those of Baseline without SEFAR.

| $L_1$ | $L_2$ | $L_3$ (KL) | $L_3$ (MSE) | ResNet-18 | ViT-S |
|---|---|---|---|---|---|
| ✓ | | | | 83.55 | 52.18 |
| ✓ | ✓ | | | 84.73 | 53.98 |
| ✓ | ✓ | ✓ | | **86.05** | 55.59 |
| ✓ | ✓ | | ✓ | 85.86 | **55.88** |

Table 4: Ablation study on which component in SEFAR works and whether SEFAR can work on Vision Transformer. ignifies that this loss term is retained during training, while a blank space indicates the removal of this term. $L_3$ (KL) means that distillation uses the KL divergence between the predictions of sparse features and original features. $L_3$ (MSE) means that the distillation loss is calculated using MSE distance.

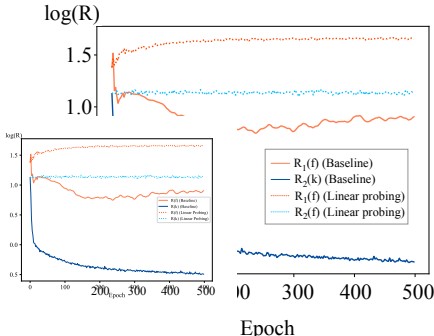

Figure 6: $R_1(\mathbf{F})$ and $R_2(\mathbf{K})$ of Baseline with SEFAR and linear probing with SEFAR, respectively. In linear probing, because the backbone is frozen, $R_1(\mathbf{F})$ and $R_2(\mathbf{K})$ do not decrease, which indicates that they cannot be minimized through the learning of the classification head. In Baseline, where the backbone is trainable, SEFAR causes the upper bound of the generalization error to decrease. This is one of the reasons that SEFAR improves the model's generalization ability.

## 5 CONCLUSION

In this paper, we introduce SEFAR to enhance the classification performance of various fine-tuning methods across five datasets. We also provide a theoretical proof for SEFAR, which introduces and minimizes an upper bound of generalization error related to Rademacher complexity in the final loss function. Due to computational limitations, future research will investigate whether SEFAR has a positive impact on larger-scale models and more types of tasks. From a theoretical perspective, we also aim to explore more properties of feature eigenvalues.

## 6 THE DERIVATION OF EQUATION 5

Since the solution of $\mathbf{w}_1$ can be obtained from Equation 4, then we have:

$$
\begin{aligned}
&\|\mathbf{w}_1^*\mathbf{F} - \mathbf{y}\|^2 + \|\mathbf{w}_1^*\mathbf{F} - \mathbf{s}\|^2 + 2\lambda_1\|\mathbf{w}_1^*\|^2 \\
&= \|\mathbf{y}\|^2 + \|\mathbf{s}\|^2 - 2(\mathbf{w}_1^*)^\top\mathbf{F}^\top\mathbf{y} - 2(\mathbf{w}_1^*)^\top\mathbf{F}^\top\mathbf{s} + 2(\mathbf{w}_1^*)^\top\mathbf{F}^\top\mathbf{F}\mathbf{w}_1^* + 2\lambda_1(\mathbf{w}_1^*)^\top\mathbf{w}_1^*, \\
&= \|\mathbf{y}\|^2 + \|\mathbf{s}\|^2 - \frac{1}{2}(\mathbf{s}+\mathbf{y})(\mathbf{F}^\top\mathbf{F} + \lambda_1\mathbf{I})^{-1}\mathbf{F}^\top\mathbf{F}(\mathbf{s}+\mathbf{y}) \\
&= \|\mathbf{y}\|^2 + \|\mathbf{s}\|^2 + \lambda_1 R(\mathbf{F}).
\end{aligned}
\tag{11}
$$

So we can get Equation 5 with Equation 6

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
