# A APPENDIX

## A.1 HYPERPARAMETER CONFIGURATION OF SEFAR

|  | EuroSAT | ISIC | ChestX | CropDisease | CIFAR-10 |
|---|---|---|---|---|---|
| Baseline | $\lambda = 5, t = 10, s = 0.3$ | $\lambda = 1, t = 1, s = 0.8$ | $\lambda = 1, t = 1, s = 0.8$ | $\lambda = 5, t = 10, s = 0.3$ | $\lambda = 5, t = 5, s = 0.3$ |
| linear probing | $\lambda = 5, t = 1, s = 0.3$ | $\lambda = 5, t = 1, s = 0.3$ | $\lambda = 5, t = 1, s = 0.3$ | $\lambda = 5, t = 5, s = 0.3$ | $\lambda = 5, t = 1, s = 0.3$ |
| L2-SP | $\lambda = 5, t = 10, s = 0.3$ | $\lambda = 0.5, t = 5, s = 0.8$ | $\lambda = 5, t = 10, s = 0.8$ | $\lambda = 5, t = 10, s = 0.3$ | $\lambda = 5, t = 5, s = 0.9$ |
| DELTA | $\lambda = 5, t = 2, s = 0.3$ | $\lambda = 5, t = 2, s = 0.3$ | $\lambda = 5, t = 2, s = 0.3$ | $\lambda = 5, t = 2, s = 0.3$ | $\lambda = 5, t = 1, s = 0.3$ |
| Surgical tuning | $\lambda = 5, t = 2, s = 0.3$ | $\lambda = 5, t = 2, s = 0.3$ | $\lambda = 5, t = 2, s = 0.3$ | $\lambda = 5, t = 2, s = 0.3$ | $\lambda = 5, t = 1, s = 0.3$ |

Table 5: SEFAR's hyperparameter configuration of Table 1. $\lambda$ represents the weight of $L_3$, $t$ means the distillation temperature coefficient and $s$ means the sparsity of $\mathbf{M}$.

|  | EuroSAT | ISIC | CropDisease | ChestX |
|---|---|---|---|---|
| Baseline | $\lambda = 5, t = 2, s = 0.6$ | $\lambda = 5, t = 2, s = 0.6$ | $\lambda = 5, t = 2, s = 0.3$ | $\lambda = 5, t = 2, s = 0.6$ |
| linear probing | $\lambda = 5, t = 2, s = 0.6$ | $\lambda = 5, t = 2, s = 0.3$ | $\lambda = 5, t = 2, s = 0.6$ | $\lambda = 5, t = 2, s = 0.6$ |

|  | EuroSAT | ISIC | CropDisease | ChestX |
|---|---|---|---|---|
| Baseline | $\lambda = 5, t = 2, s = 0.3$ | $\lambda = 5, t = 2, s = 0.3$ | $\lambda = 5, t = 2, s = 0.6$ | $\lambda = 5, t = 1, s = 0.6$ |
| linear probing | $\lambda = 5, t = 1, s = 0.8$ | $\lambda = 5, t = 2, s = 0.6$ | $\lambda = 5, t = 2, s = 0.6$ | $\lambda = 5, t = 2, s = 0.6$ |

Table 6: SEFAR's hyperparameter configuration of Table 2. Above: 5-way 1-shot. Below: 5-way 5-shot.