# OpenReview forum: "SEFAR: SparsE-FeAture-based Regularization for Fine-Tuning on Limited Downstream Data"
_ICLR.cc/2024/Conference — ICLR 2024 Conference Withdrawn Submission_

### Official Review · Reviewer_affX · 2023-10-29

**Soundness:** 3 good
**Presentation:** 2 fair
**Contribution:** 3 good
**Rating:** 5
**Confidence:** 4

**Summary:**

This paper tackles the problem of fine-tuning on limited downstream data. The authors propose to fine-tune the backbone network with two task heads, one of which uses dropout to mask random features. The produced logits from this head are then used as a supervision signal for the other head, called "self-distill". Experimental results show that this method can improve the final performance on various fine-tuning methods.

**Strengths:**

- The proposed method seems interesting and novel to me. Additionally, the method is simple and easy to implement.
- The experimental results are promising, as the proposed method shows improvement over several base methods.

**Weaknesses:**

- The proposed method introduces several hyperparameters, including sparsity, temperature, and weight, and some of them are quite sensitive. However, the authors didn't discuss how they chose the hyperparameters.
- The dataset evaluated in this paper is somewhat limited. There are numerous datasets designed for the studied scenario, such as the VTAB-1k benchmark, which contains 19 datasets. Authors may consider evaluating their method on more datasets.
- The theoretical analysis provided in this paper seems vague to me. Please refer to the related part in my questions.

**Questions:**

- It is interesting to see in Fig. 3(a) that even when the sparsity is 0.1, the method still shows obvious improvement. What would happen if the sparsity is set to 0?
- The paper provides an analysis to show that SEFAR reduces the upper bound of generalization error. However, it is hard for me to understand why the introduced sparse-feature-based regularization helps. How does the sparsity of features affect generalization? It seems that the paper should discuss this more rigorously.
- The paper studies the situation where limited downstream data are provided, and the training set only contains 200 samples. What would happen if the training set is increased? Is there any further explanation?

---

> ### Author Response · Authors · 2023-11-22
> **Response to Reviewer affX**
>
> We really appreciate your comments and reviewing. We supply some experiments for this response.
>
> **Q1. How do you choose hyperparameters?**
>
> **A1.** We have $\lambda$ (the weight of $L_3$), $t$ (the distillation temperature coefficient) and $s$ (the sparsity of the binary mask). We choose the optimal hyperparameters by a grid search. The detailed configuration of each experiment is shown in our appendix.
>
> **Q2. The evaluation dataset is limited.**
>
> **A2.** We conduct experiments on more datasets with ResNet-18:
>
> |           | Caltech101 | CIFAR100 | FLowers | Oxford | SVHN  | DTD   |
> |-----------|------------|----------|---------|--------|-------|-------|
> | w/o SEFAR | 57.85      | 15.06    | 41.54   | 39.30  | 73.90 | 26.29 |
> | w/ SEFAR  | 59.09      | 16.57    | 42.82   | 41.47  | 76.62 | 29.15 |
>
> In this table, SEFAR shows its consistent positive gain for the full fine-tuning method.
>
> **Q3. What if sparsity is set to 0 (figure 3a)**
>
> **A3.** When the sparsity is set to 0, on the EuroSAT dataset, the results with different distillation temperature coefficients are:
>
> | Method                  | Accuracy |
> |-------------------------|----------|
> | w/o SEFAR               | 83.55    |
> | w/ SEFAR, s=0, t = 1    | 83.52    |
> | w/ SEFAR, s=0, t = 2    | 84.61    |
> | w/ SEFAR, s=0, t = 5    | 84.46    |
> | w/ SEFAR, s=0, t = 10   | 84.07    |
> | w/ SEFAR, s=0.3, t = 10 | 86.05    |
>
> Results with s=0 are not as good as the result with s=0.3, which illustrates that the sparsity is important.
>
> **Q4. How does the sparsity of features affect generalization?**
>
> **A4.** Figure 3 (a) provides an initial insight into the impact of sparsity on linear probing. In our experiments, we conducted a grid search across values {0.3, 0.6, 0.8, 0.9} to identify the optimal sparsity value. In the future, we may explore the detailed mechanism.
>
> **Q5. What would happen if the training set is increased?**
>
> **A5.** We conduct experiments with various training dataset sizes and the results are:
> |           | 20 images | 200 images | 500 images | 2000 images |
> |-----------|:---------:|:----------:|:----------:|:-----------:|
> | w/o SEFAR |   43.65   |    83.55   |    89.65   |    95.06    |
> |  w/ SEFAR |   48.03   |    86.05   |    90.87   |    95.26    |
>
> The gain in performance with SEFAR becomes more pronounced as the number of training set images decreases. Additionally, as the number of training images increases, the additional generalization can be derived from the incremental data. Therefore, SEFAR is particularly suitable for use when fine-tuning with a limited number of images.

---

### Official Review · Reviewer_Bx9c · 2023-10-30

**Soundness:** 2 fair
**Presentation:** 3 good
**Contribution:** 1 poor
**Rating:** 3
**Confidence:** 3

**Summary:**

This paper introduces a simple regularization method called SparsE-FeAture-based Regularization (SEFAR) for model fine-tuning, particularly when a limited amount of downstream data is available. The proposes SEFAR is simple and easy to implement on pre-trained models. The empirical experiments show it can achieves good results on down-streaming tasks with limited target data.

**Strengths:**

1. The proposed method is very simple and easy to understand.
1. The experiment results show that the proposed SEFAR enhances the performance of various fine-tuning methods.

**Weaknesses:**

1. The proposed method is kind of heuristic. It is actually an extension of dropout regularization (L_1 + L_2).  The self-distillation loss (either the CE loss or  MSE loss) is also a popular trick in machine learning tasks.  Therefore, the technical contribution of this paper is very limited.
1. The theoretical analysis is proved for an oversimplified scenario: one-class (output) case with the MSE loss, which is very different from scenario (multi-class classification setting) studied in this paper.
1. The organization of this paper can be improved, e.g., Algorithm 1, Table 4, Figure 6.

**Questions:**

1. It is very strange that the t-SNE of each class in Fig.2c forms as disk. How to understand that ``randomly dropping some dimensions to generate sparse features may result in enhanced discriminative capabilities''? It seems interesting.
1. Why the theoretical analysis is put in the experiment section. It is very strange.
1. How Eq.3 is derived based on the proposed method in Eq.2?  Why it is a reasonable approximation.

---

> ### Author Response · Authors · 2023-11-22
> **Response to Reviewer Bx9c**
>
> Thanks for your comments. Here are our responses.
>
> **Q1. Dropout and Self-distillation are popular. So the contribution is limited.**
>
> **A1.** Our main contribution is we discover that the sparse feature can be a good teacher in the self-distillation process. We theoretically and experimentally prove why it can work well.
>
>
> **Q2. The theoretical analysis (one-class case) with MSE loss is oversimplified.**
>
> **A2.** We think that one-class case + MSE Loss analysis is a common way, as shown in [1]. We also empirically confirmed that the results of MSE loss are consistent with the cross-entropy case in practice, which can be observed in Table 4 of our submitted script.
>
> **Q3. t is very strange that the t-SNE of each class in Fig.2c forms as disk. How to understand that ``randomly dropping some dimensions to generate sparse features may result in enhanced discriminative capabilities''? It seems interesting.**
>
> **A3.** We also consider this to be a fascinating phenomenon: when randomly masking certain features, the performance of feature clustering improves. In the future, we plan to conduct more experiments and theoretical exploration to understand the underlying reasons.
>
> **Q4. The organization of experiment section.**
>
> **A4.** We think it is a good suggestion and we may reorganize the paper by add a new section for theoretical analysis.
>
> **Q5. How Eq. 3 is derived based on the proposed method in Eq.2?**
>
> **A5.** Please refer to **A2**.
>
> **Reference**
>
> [1] Hossein Mobahi, Mehrdad Farajtabar, Peter L. Bartlett. Self-Distillation Amplifies Regularization in Hilbert Space. In NeurIPS 2020.

---

### Official Review · Reviewer_6bB7 · 2023-11-01

**Soundness:** 1 poor
**Presentation:** 2 fair
**Contribution:** 1 poor
**Rating:** 3
**Confidence:** 4

**Summary:**

The author claims to introduce a regularization method for limited data classification finetuning problem. The method is pluggable onto other finetuning methods. The method is to add another classification head that 1) has a mask on input to mask out some input features; 2) its prediction distills the original classification head using all features. This paper also provides some theoretical explanation with approximation about why this method could bring additional improvement. In addition, the author provides experiments on several datasets, compared with several finetuning methods.

**Strengths:**

-This paper try to tackle an important problem: finetuning with limited data. The experimental results show some improvement over several other methods.
-The paper's writing overall is clear. But there are some important details missing.
-There are quite some ablation study done. The experiments are over multiple dataset and multiple baseline methods, with proper confidence level stated. However the experiments are not well aligned with authors' claims.

**Weaknesses:**

Originality

This paper states the main contribution in terms of method originality to be “additional classification and self-distillation loss function computed using the sparse features”. The method of additional classification head and self distillation from the additional head to the original head is a very common technique used in the community. Itself doesn’t have any novelty.
The main difference here in my opinion is that the authors used a mask on the inputs of the additional head: I personally do not know papers with similar methods. However it is very unclear how the mask is generated (the maskgen algorithm mentioned): from the limited information that I can find in the paper, it seems that the mask is randomly generated, not learned or anything. In this sense, basically this method does a random dropout on a certain layer (i.e. the input of the additional head). It is not surprising that it can bring a regularization effect. The originality of this method seems weak to me.
However, sometimes simple combination of existing methods may have a surprising effect, if the empirical study strongly supports it. Unfortunately as mentioned below in quality section, I don’t think the empirical study is convincing enough.

Quality

- 4.2 ablation study: what about the impact of mask in additional head? I.e. If we have L1,L2,L3 and no mask (i.e. all feature used; or sparsity=0), what is the result? what about we have self-distillation and just apply dropout (on some layers or just all task heads)? Would we observe similar improvement as the proposed method?

Clarity

- The maskgen algorithm is not properly described. Is it just randomly generates a mask with a given sparsity? How many sparsity parameters you tuned for the experiments given? (You mentioned that the sparsity has minimal impact on W2 in table 3. This is not what I would like to ask). My question is for major comparison table 1, for each row w/ SEFAR, how many sparsity parameters you have tuned? And what is their impact on the classification results (of W1)? An example is your figure 3(a).

Significance

The author claims that this paper targets to help finetuning problem with limited data. However I do not see any explanation or empirical study to support this.

-	Does SEFAR helps less with finetuning problem with more data? Is there any reason theatrically? Or empirical evidence?

-	What is considered limited data? Why there is no related work or empirical study comparing with methods that also specifically address limited data finetuning methods?

It also claims that this method “significantly enhance the performance of any fine-tuning method”. However, I see that the authors choose to compare with only several finetuning methods that are not claimed to be state-of-the-art (linear probing, L2-SP (Xuhong et al., 2018), DELTA (Li et al., 2019), and surgical fine-tuning (Lee et al., 2022)). To support such claim that the authors made, experiments with more recent SOTA methods, and on the datasets that those papers used are necessary.


------------------
after seeing author's response:
Thank you authors for clarifying and adding additional ablation study that I asked. It does help improve the confidence level of the effectiveness of the proposed method on the given datasets. However my major concern on the significance section was not fully addressed. I still think the claims made (e.g. any finetuning method, limited data) are not supported. I keep my original score.

**Questions:**

4.2 ablation study what dataset? Is it EuroSAT?

---

> ### Author Response · Authors · 2023-11-22
> **Response to Reviewer 6bB7**
>
> Thanks for your reviewing. Let's answer your comments as possible as we can.
>
> **Q1. For the maskgen algorithm and the way to generating the sparse mask.**
>
> **A1.** In this paper, we clarify that the sparse mask is randomly generated, without any learning process. Specifically, we generate an all-one matrix with the same shape of the feature. Then we randomly select a ratio of elements and set them 0. The ratio is the sparsity mentioned as $\lambda$.
>
> **Q2. If we have $L_1$, $L_2$, $L_3$ and no mask (i.e. all feature used; or sparsity=0), what is the result.**
>
> **A2.** Here are the results on EuroSAT dataset:
>
> | Method                           | Accuracy |
> |----------------------------------|----------|
> | w/o SEFAR                        | 83.55    |
> | w/ SEFAR (sparsity s= 0, t=1)    | 83.52    |
> | w/ SEFAR (sparsity s= 0, t=2)    | 84.81    |
> | w/ SEFAR (sparsity s= 0, t=5)    | 84.56    |
> | w/ SEFAR (sparsity s= 0, t=10)   | 84.27    |
> | w/ SEFAR (sparsity s= 0.3, t=10) | 86.05    |
>
> We also explore if L2 itself can work well on EuroSAT, the results are:
>
> | Backbone                     | Loss function        | Accuracy |
> |------------------------------|----------------------|----------|
> |                              | L1 only              | 83.55    |
> | ResNet-18                    | L2 only (DropOut)    | 85.10    |
> |                              | L1+L2+L3 (Our SEFAR) | 86.05    |
> |                              | L1 only              | 52.18    |
> | ViT-small (Full fine-tuning) | L2 only (DropOut)    | 53.87    |
> |                              | L1+L2+L3 (Our SEFAR) | 55.59    |
>
> According to this table, we can observe that although SEFAR seems to be a combination of DropOut and self-distillation in its implementation, the gains in terms of results are even greater, especially on ViT-small.
>
> **Q3. Is ablation study in 4.2 conducted on EuroSAT?**
>
> **A3.** Yes, it is. And the experiment configuration is the same.

---

### Official Review · Reviewer_f9tb · 2023-11-09

**Soundness:** 2 fair
**Presentation:** 3 good
**Contribution:** 2 fair
**Rating:** 3
**Confidence:** 4

**Summary:**

In this paper, the authors propose a spare-feature-based regularization method for improving the fine-tuning efficiency. Their key idea is to use the prediction results based on the generated sparse features to distill the prediction results with the complete features. They finally formulate it into two regularizations for fine-tuning.  They evaluate the performance of their proposed method by conducting experiments on some datasets.

**Strengths:**

1. This paper is well-written and easy to read.
2. The experimental results imply the proposed method is technically correct.
3. The proposed method is easy to implements.

**Weaknesses:**

1.	The authors conduct experiments on ResNet18 with ImageNet. It is insufficient. They are recommended to evaluate their proposed method on the language models and other larger CNN based models.
2.	It is unclear to me whether different masks can lead to different performance and how to choose a good mask for the proposed method. Can we optimize the masks during finetuning to improve the proposed method?
3.	The results in Figure 2 shows that the improvement achieved by the proposed method is marginal.

**Questions:**

Please refer to the comments about the weaknesses.

---

> ### Author Response · Authors · 2023-11-22
> **Response to Reviewer f9tb**
>
> Thank you very much for your review and suggestions. Our response to your inquiries is as follows:
>
> **Q1. More evaluation on language models and other larger CNN models**
>
> **A1.** Due to constraints in time and computational resources, we conducted experiments using VGG11, VGG11 (with BN), MobileNet v2, LoRA+ViT-B, and LoRA+ViT-L on the EuroSAT dataset. The results are as follows:
>
>
> | Backbone        | Method     | Accuracy |
> |-----------------|------------|----------|
> |                 | w/o SEFAR  | 72.86    |
> | VGG11           | w/ SEFAR   | 77.11    |
> |                 | $\Delta$   | +4.25    |
> |                 | w/o SEFAR  | 84.03    |
> | VGG11 (with BN) | w/ SEFAR   | 85.58    |
> |                 | $ \Delta$  | +1.55    |
> |                 | w/o SEFAR  | 86.19    |
> | MobileNet v2    | w/ SEFAR   | 87.74    |
> |                 | $ \Delta$  | +1.55    |
> |                 | w/o SEFAR  | 88.54    |
> | LoRA+ViT-B      | w/ SEFAR   | 91.37    |
> |                 | $ \Delta$  | +2.83    |
> |                 | w/o SEFAR  | 89.96    |
> | VGG11           | w/ SEFAR   | 92.76    |
> |                 | $ \Delta $ | +2.80    |
>
> It is obvious that SEFAR can work across different network families and scales.
>
> **Q2. Whether different masks can lead to different performance and how to choose a good mask. Can we optimize the masks during finetuning.**
>
> **A2.**  In our approach, we introduce a randomly generated binary mask, where the sparsity (i.e., the proportion of zeros) is the only variable that needs to be controlled. During the training process, the mask is randomly generated and does not need to be optimized.
>
> **Q3. T-SNE results show marginal improvement.**
>
> **A3.** In the T-SNE results, our primary focus is to highlight the differences in clustering outcomes between the original features and sparse features. Specifically, in scenarios with a limited number of training samples, sparse features exhibit superior clustering performance. Therefore, utilizing the classification results generated by sparse features as the teacher in self-distillation proves to be more advantageous.